# Sleep and Social Wellness: Does Current Subjective and Objective Sleep Inform Future Social Well-Being?

**DOI:** 10.3390/ijerph191811668

**Published:** 2022-09-16

**Authors:** Sarah M. Ghose, Morgan P. Reid, Natalie D. Dautovich, Joseph M. Dzierzewski

**Affiliations:** Department of Psychology, Virginia Commonwealth University, Richmond, VA 23284, USA

**Keywords:** actigraphy, sleep diary, sleep quality, social well-being, PSQI, longitudinal

## Abstract

Objectives: The present study aimed to investigate the link between sleep and broader social well-being. Specifically, the current study evaluated whether subjective and objective sleep indices were associated with subsequent social well-being. Methods: The archival data from the Midlife in the United States Study (MIDUS II and III, Project 1 and 4) were utilized for the current investigation. The participants completed cross-sectional surveys as part of their involvement in both study waves, 10 years apart. They were 213 adults, 59.6% female-identifying, with an average age of 56 years, who completed 8 days of sleep measurement via wrist actigraphy, sleep diary, as well as the PSQI. The participants also completed the measures of depressive symptoms and social well-being. Results: The actigraphic total sleep time, the diary-reported sleep quality, and the global sleep quality measured by the PSQI emerged as the significant predictors of social well-being over a 10-year period. Conclusions: The present study is an initial step in providing evidence for the importance of sleep for social functioning. Future research should attend to the association between past sleep behaviors and social functioning, specifically the mechanisms by which sleep is associated with social well-being as well as the temporal associations in an adult sample.

## 1. Sleep and Social Wellness: Does Current Subjective and Objective Sleep Inform Future Social Well-Being?

As the events of the past few years have revealed, our connections to others are important for our overall well-being. Social well-being, the extent to which someone can navigate society and experience belongingness, is critical for inter- and intrapersonal wellness, including the maintenance of health, disease prevention, and improved access to health resources [1]. Despite the broad implications of social well-being, less is known about the contributing role of daily health behaviors such as sleep. Sleep quantity and quality contribute to cognitive, physical, mental health, and interpersonal functioning [2,3,4]. However, the associations between sleep and general social well-being remain largely unknown. The extent to which poor sleep may be tied to decreased social functioning over time requires further investigation. Given that sleep is a daily, salient, and highly modifiable behavior, it may serve as a target for promoting and protecting social well-being. As such, the current study focuses on social well-being as a correlate of objectively and subjectively measured sleep. 

### 1.1. Social Well-Being

According to the biopsychosocial model of health, an individual’s wellness extends across the biological, psychological, and social domains [5]. Despite the prioritization of biological and psychological factors in the study and treatment of health outcomes, individuals exist within the social environment, and their interactions within that environment need to be considered. Wellness across the biological, psychological, and social domains is necessary for one to “flourish”, rather than “languish” [6]. The existing research on social health has largely attended to societal structures, including social supports, living environments, and specialized psychoeducational programs. However, social health consists of both societal structures and individual social well-being [5]. 

Social well-being, encompassing the social expectations of the self and the society at large, is the “appraisal of one’s circumstance and functioning in society” [5]. According to Keyes [5,6], social well-being dimensions include coherence, actualization, integration, acceptance, and contribution. Overall, social well-being represents an individual’s assessment of their ability to navigate society as well as the extent to which they experience societal belonging. Social well-being and its counterpart, isolation, have emerged as important determinants of health outcomes such as increased inflammation [7], heightened blood pressure [8], and increased disease susceptibility [9]—outcomes that are financially costly to both the individual and the healthcare system broadly [10]. The high personal and societal costs associated with these health outcomes point to the importance of increased empirical attention to the mechanisms underlying social well-being. Sleep is a universal, modifiable mechanism that has been readily revealed in the literature to contribute to two components of Keyes’ [6] flourishing: psychological and emotional [2,3,4]. Social well-being, the third component of flourishing, may be another domain in which sleep is linked.

### 1.2. Sleep and Social Well-Being

Sleep is a health outcome that has been tied to the components of social well-being. Most research in this area has focused on the association between social isolation and sleep-related outcomes, such as disturbed sleep, insomnia, and daytime fatigue, both cross-sectionally and longitudinally [11,12]. Although the links between social well-being and sleep outcomes have been established, a growing body of literature has shifted focus toward the reciprocal association—the role of sleep for social well-being. 

A recent body of work suggests that poor sleep is associated with a range of maladaptive interpersonal consequences, including increased interpersonal conflict [13], social withdrawal [14,15], and decreased accuracy in the identification and interpretation of facially communicated emotions essential to empathy and affiliation [16] following a night of inadequate sleep. Sleep efficiency is also associated with interpersonal functioning, as greater sleep efficiency predicted less negative partner interaction the following day in heterosexual couples [17]. Beyond interpersonal consequences, cross-sectional studies have also demonstrated an association between sleep duration and an individual’s sense of belongingness, trust, and social engagement. Both long and short sleepers show deficits in social capital compared with those sleeping 7–8 h, with long sleepers less likely to report social memberships, and short sleepers reporting significantly less helping behaviors [18]. Further, the individuals with moderate-to-severe insomnia report significantly lower levels of trust for individuals in their neighborhoods, alongside decreased feelings of community belongingness. Similarly, severe insomnia contributes to an increased “disbelief in a just world” among older adults [12]. Indeed, many dimensions of sleep health, including duration, efficiency, quality, and insomnia have been implicated in the association between sleep and the discrete elements of social well-being. 

As an example of the mechanisms tying poor sleep to social well-being, Simon and Walker [15] provide a framework for understanding how poor sleep may contribute to both loneliness and social withdrawal. Sleep-deprived individuals have higher levels of mistrust and view others as less socially desirable; additionally, others are more likely to disengage from poorly rested individuals [14], resulting in a bidirectional cycle. In other words, poor sleep and poor social well-being create a two-way street. 

Although there is benefit in understanding the bidirectional associations between sleep and social well-being, the present study assesses the longer-term associations between sleep and social well-being. The previous studies examining sleep and social well-being have induced sleep deprivation in the laboratory setting [14] or gathered the sleep data solely through daily diaries [13] or self-reported measures [17]. Although self-reported data provide valuable information about the perceptions of sleep, there are discrepancies between the self-reported and objective measures of sleep [19], with education level and sleep-related perceptions posited as potential exacerbating factors [20,21]. Therefore, the current study aimed to investigate sleep variables across objective and subjective modalities (actigraphy, sleep diary) to identify the facets of sleep that are most associated with social well-being. Furthermore, actigraphy and sleep diaries only captured one week of the participants’ sleep, which may not be representative of their typical sleep patterns; therefore, a global self-reported measure of habitual sleep was also examined. 

### 1.3. The Current Study 

In the present study, we examined the extent to which objectively and subjectively measured sleep predicts social well-being in adults. We hypothesized that a higher amount of actigraphic total sleep time [18] and sleep efficiency [17], better sleep-diary-reported sleep quality [13], shorter sleep-onset latency [13], and lower global sleep disturbance [13] will be associated longitudinally with better social well-being. 

## 2. Methods

### 2.1. Participants

Secondary data analyses were conducted utilizing the data from the Midlife in the United States (MIDUS) study. The current project examined the data from the MIDUS II biomarker and MIDUS III projects. MIDUS II was conducted from 2004 to 2006, and MIDUS III was conducted from 2013 to 2014.

Although a total of *N* = 2653 individuals participated across the MIDUS II biomarker and MIDUS III projects, the current study retained only those participants with full information on target measures. As such, the final sample utilized in the current study included 213 participants. The participants ranged in age from 35 to 83 years of age (*M* = 55.68, *SD* = 10.92), were predominantly female (59.6%), college-educated (51.7% reported some type of college degree), and self-identified as white (95.3%). Further, the sample endorsed higher levels of depressive symptomatology, *M* = 32.54, on the Center for Epidemiologic Studies Depression Scale (CES-D; [22]). Using a cut-off of 5 on the Pittsburgh Quality Sleep Index (PSQI; [23]), 61.5% of the sample were good sleepers (<5), and 38.5% of the sample were poor sleepers (>6). See Table 1 for participant characteristics. 

### 2.2. Procedure

The MIDUS II biomarker and MIDUS III data collections were reviewed and approved by (1) the Education and Social/Behavioral Sciences and (2) the Health Sciences at the University of Wisconsin-Madison. After providing informed consent to participate in the study procedures, the MIDUS II, Project 4 participants engaged in a protocol that included wearing an actigraphic device and a Mini Mitter-64 activity monitor (see below) and completing a sleep diary for 8 days. They also completed a survey about their sleep. As part of their participation in MIDUS III, the participants completed psychosocial surveys that included measures of depressive symptomatology and social well-being. More information on participant recruitment and study design is available at the official MIDUS website.

### 2.3. Measures

#### 2.3.1. Demographic and Health Information

The present study utilized the participants’ age, gender, income, psychosocial, and sleep data.

#### 2.3.2. Depressive Symptoms

The associations between psychological distress, sleep, and social wellness are well-documented [23,24]. The Center for Epidemiologic Studies Depression Scale (CES-D) is a 20-item self-reported measure designed to assess respondents’ depressive symptom frequencies over the past week [22]. The present study utilized a 19-item measure, as the sleep-related item was removed. Responses are made on a 4-point Likert scale (from 0 = rarely or none of the time to 3 = most or all of the time), with total scores ranging from 0 to 60. The scale includes such items as, “I was bothered by things that usually don’t bother me.” Higher scores indicate increased depressive symptomatology. The CES-D has been shown to be a reliable and valid measure [22]. 

#### 2.3.3. Actigraphy

The actigraphic device model utilized in MIDUS II, Project 4 was a Mini Mitter-64 activity monitor. This device is worn on the non-dominant wrist and, based on activity movements, determines sleep or wake states every 30 s. Actigraphy has been shown to be a reliable, convenient alternative to the polysomnographic measures of sleep for clinical trials and multi-measure studies [25]. Sleep was assessed across 8 consecutive days. Actiware 5.0 was used to calculate the following variables: sleep-onset latency (SOL: the time from lights out to the first sleep onset); the total sleep time (TST); and sleep efficiency (SE: the ratio of the time spent asleep to the total time spent in bed). 

#### 2.3.4. Sleep Diary

The Pittsburgh Sleep Diary (PghSD) is a daily self-reported instrument in which participants report their bedtime and wake time, in addition to their use of prescribed and over-the-counter-medication, napping behavior, caffeine consumption, number of awakenings, minutes to sleep onset, and sleep quality [26]. The current study participants completed sleep diaries for 8 days, immediately before they went to bed and after they woke up for the day. The diary variables of interest to the present study were SOL and sleep quality (SQ: perceived sleep health). The PghSD has been shown to be a reliable and valid measure of sleep [25]. 

#### 2.3.5. Sleep Quality

The Pittsburgh Sleep Quality Index (PSQI) is a 19-item self-reported questionnaire designed to measure respondents’ sleep quality over the past month. Respondent scores across seven components, namely, subjective sleep quality, sleep latency, sleep duration, sleep efficiency, sleep disturbance, use of medication, and daytime dysfunction, are aggregated to arrive at a single global score. Responses are made on a 4-point Likert scale to such items as, “During the past month, how many hours of actual sleep did you get at night?” Lower scores are indicative of better sleep. The PSQI has been shown to be a reliable and valid measure of sleep quality [27,28,29].

#### 2.3.6. Social Well-Being

The Social Well-Being Scale is a measure of five dimensions of social well-being: social coherence, social integration, social acceptance, social contribution, and social actualization [5,6]. The overall social well-being was calculated as an aggregate of the participants’ five subscale scores. This scale was administered to the participants during their participation in MIDUS III. Survey responses are made on a 7-point Likert scale (from 1 = strongly agree to 7 = strongly disagree) with total scores ranging from 14 to 98. The scale includes such items as “I have something valuable to give to the world” and “I believe that people are kind.” Higher scores are indicative of higher levels of social well-being. The Social Well-Being subscales are generally considered valid and reliable measures [5]. In the present sample, reliability values for all the five social well-being subscales were as follows: coherence (α = 0.23); integration (α = 0.70); acceptance (α = 0.37); contribution (α = 0.23); actualization (α = 0.35). The reliability of the summated scale in the current sample was α = 0.41. The low reliability of these scales in the sample may reflect (1) the limited number of items per subscale (e.g., 3 items per subscale); (2) the multifaceted nature of social well-being; or (3) the reality that individuals may differ on their endorsements of specific aspects of social well-being over others.

#### 2.3.7. Statistical Analyses

All statistical analyses were conducted with IBM SPSS Version 26 software. The study hypotheses were tested using multiple hierarchical regression analyses to determine the longitudinal associations of sleep variables at Time 1 with social well-being at Time 2. Age, sex, depressive symptomatology, and household income were utilized as covariates in regression analyses with the actigraphic, sleep diary, or PSQI sleep quality variables entered in step 2. Separate multiple hierarchical regression analyses were conducted to assess the differential amount of variance in social well-being accounted for by the objective (TST, SOL, SE), self-reported (SOL, SQ), and global (PSQI) sleep indices. Power analysis using G*Power31 showed that for multiple regression analysis with 7 predictors (the largest model), a sample size of at least 153 is needed to predict an R2 of at least 0.15 and a power level of 0.95, suggesting that the present study is adequately powered.

Notably, similar sleep variables were chosen for the objective and subjective measures where possible (e.g., actigraphic and diary-reported SOL). However, the total sleep time and sleep efficiency were only measured via actigraphy. The actigraphic and diary sleep variables were grouped and entered simultaneously within their respective analyses to ascertain which of these multiple variables was most associated with social well-being while minimizing Type I error and accounting for the smaller group sample sizes.

## 3. Results

### 3.1. Demographic and Health Covariates 

Household income emerged as a significant predictor of social well-being across all the study analyses, all *p*s < 0.001, whereby higher household income was positively associated with higher social well-being. Higher levels of depressive symptoms were significantly associated with decreased social well-being when included in step 2 of the model with actigraphic indices of sleep (TST, SOL, SE). Depressive symptoms did not predict social well-being in either the diary-reported or global sleep analyses. Across all the analyses, age and sex did not surface as significant predictors of social well-being. See Table 2 for bivariate correlations and descriptive statistics.

### 3.2. Actigraphic Indices

The total sleep time at time 1 was significantly associated with social well-being at time 2, *R^2^* = 0.11, *F*(7, 205) = 3.70, whereby a lower amount of total sleep time predicted higher social well-being over time (*β* = −0.23, *p* < 0.05). The actigraphically measured SE and SOL were not significantly associated with social well-being (see Table 3).

### 3.3. Diary-Reported Indices 

The diary-reported SQ was a significant predictor of social well-being longitudinally, *R*^2^ = 0.12, *F*(6, 203) = 4.49, whereby a higher SQ was associated with better social well-being (*β* = −0.24, *p* < 0.001). The self-reported SOL was not significantly associated with social well-being (see Table 4).

### 3.4. PSQI

Global sleep quality was associated with social well-being longitudinally, *R*^2^ = 0.09, *F*(5, 207) = 3.87, whereby worse sleep quality predicted lower social well-being (*β* = −0.15, *p* < 0.05; see Table 5). 

Global sleep quality, dichotomized at a cut point of 5, 0 = good sleepers and 1 = poor sleepers [23], was associated with social well-being longitudinally, *R*^2^ = 0.09, *F*(5, 207) = 3.84, *p* < 0.05. Those in the poor sleeper group evidenced significantly worse social well-being (*β* = −0.15, *p* < 0.05; see Table 5).

### 3.5. Post Hoc Analyses

#### Actigraphic Indices

Post hoc regression analyses were conducted to evaluate a possible U-shaped relationship between total sleep time and social well-being. The total sleep time was squared and entered into a regression analysis in Block 2, with the covariates (age, sex, income, and depressive symptoms) in Block 1. The results of this analysis were consistent with the results of our initial analysis of the association between total sleep time and social well-being. Specifically, the total sleep time^2^ was inversely associated with social well-being, *R*^2^ = 0.11, *F* (5, 204) = 5.03, whereby an increased amount of total sleep time^2^ predicted lower social well-being, b = −0.23, *p* < 0.05. Further, depressive symptoms were inversely associated with social well-being, b = −0.18, *p* < 0.05, consistent with our initial analyses.

Post hoc regression analyses were also conducted to determine which subscale components of social well-being were significantly associated with the actigraphically measured total sleep time. The total sleep time was significantly associated with social contribution (*b* = −0.22, *p* < 0.05) and coherence (*b* = −0.23, *p* < 0.05) subscales, whereby a lower amount of total sleep time predicted higher perceptions of societal contribution and coherence, respectively. The total sleep time did not surface as a significant predictor of actualization, acceptance, or integration subscales.

## 4. Discussion

The current study aimed to build upon the previous findings that support the association between poor sleep and an array of poor social functioning outcomes, with the primary objective to determine the associations between the objective and self-reported sleep indices and global social well-being.

The results suggest that some components of sleep may be more strongly tied to social well-being than others. Sleep quality, as measured nightly via a sleep diary and globally via the PSQI, was positively associated with social well-being. This finding is consistent with previous research showing that poor sleep duration and quality, and their related daytime consequences are associated with negative interpersonal consequences [13,16]. The current results suggest that the associations between poor sleep quality and more global social well-being are significant, even across a 10-year time period. Furthermore, the importance of sleep quality, measured in the current study both globally via the PSQI and prospectively via sleep diaries, echoes other findings highlighting the important role of subjective perceptions of sleep [7]. It is possible that poor sleep quality over time may reduce the overall social well-being via the mechanisms explored in earlier studies (e.g., lower empathetic sensitivity, greater interpersonal conflict; [13,30]). Importantly, although the longitudinal associations between the diary-reported and global sleep quality and social well-being were significant, the magnitudes of these associations were not large. An increase in sleep quality was associated with a 0.25 point increase in social well-being for daily sleep quality and a 0.15 point increase for global sleep quality. 

The actigraphic total sleep time also significantly predicted social well-being. However, this association was in an unexpected direction, as a lower amount of sleep time predicted greater social well-being over time. In particular, a one-minute increase in the total sleep time was associated with a quarter-of-a-point decrease in social well-being. The current understanding in the literature is that sleep duration has a U-shaped relationship with health, and there is an optimal average duration for health benefits [31]. Notably, the post hoc regression analysis of the potential quadratic effects of total sleep time and social well-being provided the same results, namely that the total sleep time^2^ was inversely associated with social well-being. These results suggest that this association between total sleep time and social well-being may exist for individuals across the sleep duration spectrum. In light of the counterintuitive nature of this finding alongside the U-shaped relationship between sleep and health outcomes, the current finding of a negative association between total sleep time and social well-being warrants further research. Depressive symptoms served as a predictor of social well-being in combination with actigraphic sleep indices. Therefore, actigraphy may best capture the sleep-related symptoms of depression.

The post hoc analyses on the potential differential associations between total sleep time and the different social well-being components showed that sleep duration was associated with two of the five subscales—social coherence and contribution. These results suggest that total sleep time is particularly important for an individual’s perceptions of society’s overall coherence as well as their own societal contributions. Total sleep time did not evidence significant associations with acceptance, integration, and actualization. Thus, sleep behavior may be more important for more collectivistic, rather than individualistic, social well-being.

Several sleep indices were not significantly associated with social well-being, including the actigraphic sleep efficiency and the sleep-onset latency as measured by both actigraphy and sleep diaries. Nonetheless, sleep quality, which is rooted more in people’s self-perceptions of sleep rather than objective sleep parameters, was more closely associated with social well-being. Previous research has found that psychosocial characteristics affect self-reporting accuracy; those with less social support and greater depressive symptoms overreport their sleep concerns when compared with objective sleep measures [31]. Thus, these findings may suggest that sleep quality is more strongly related to social well-being because of its underlying psychosocial characteristics. 

## 5. Implications

The current study provided novel contributions to the research on sleep and social well-being in several ways. First, the inclusion of both actigraphic and subjective sleep measures provides a comprehensive analysis of multiple components of sleep. Additionally, poor sleep, particularly low-perceived sleep quality, was associated with poorer social well-being over a 10-year time period. Finally, through the use of a global measure of social well-being, we can extend the current understanding of the association between sleep and aspects of social experience to the broader social experience. 

Beyond its theoretical contributions to the literature, this work also highlights important clinical applications to consider. Healthcare providers should assess sleep in those patients with social difficulties or low social support. Additionally, as the perceived sleep quality had the strongest positive association with social well-being, sleep-related psychoeducation and cognitive restructuring (i.e., that not every night will be perfect) may be helpful in reframing patients’ sleep expectations. Particularly for older adults, regulating sleep patterns, limiting sleep medication usage, and minimizing pre-sleep cognitive activity can be important target sleep health behaviors [32].

## 6. Limitations and Future Directions

The current investigation was limited in that sleep was assessed at a single time point, which prohibited an investigation of changes in sleep. Future research should collect data on both sleep and social well-being at multiple time points. Additionally, given the smaller sample size and the stability of Time 1 and Time 2 social well-being (*r* = 0.64), we did not control for current social well-being. Future studies with a larger sample could probe the longitudinal links while adjusting for current well-being. Only the sleep parameters that appeared most commonly in the literature on sleep and social well-being were examined; however, other sleep parameters such as nighttime awakenings, daytime fatigue, or peak and trough of 24 h activity rhythms, could be associated with social well-being. Future studies should incorporate a more comprehensive battery of sleep measures in order to assess each of these dimensions of sleep health.

Additionally, the current study examined the association between sleep and social well-being outcomes over the course of 10 years. Although this is an important first step in determining the longitudinal association between these variables, the mechanisms underlying this association remain unclear. The collection of psychosocial and sleep indices at shorter intervals of time will allow for increased ability to assess the potential cumulative and/or unique contributions of sleep on social well-being longitudinally.

A strength of the present study is its inclusion of various types of sleep data, each with strengths and limitations. Although the actigraphic data were gathered across eight consecutive days, it is possible that these eight days were not representative of participants’ typical sleep. Further, actigraphy is believed to be a more accurate estimate of sleep-onset latency than a sleep diary due to the possibility of recall bias in self-reports [25]. Similarly, the sleep duration as assessed by the PSQI tends to be overinflated when compared with the actigraphic data [33,34]. It is worth noting that the lack of available data on primary sleep data disorders may have an impact on the actigraphic and social well-being data interpretation in the present study. Indeed, sleep disorder diagnoses are broadly known to have a potential impact on actigraphic data collection and social functioning. Our results should be interpreted with attention to this limitation. Similarly, controlling for current health conditions is recommended in future studies when data are available. Future studies should continue to parse the associations between data collection methods, specific sleep components and covariates, and social well-being.

Social well-being was collected via five separate self-reported indices of social well-being, each represented by scales composed of only three items. As our post hoc analyses provided evidence that sleep duration is associated with social coherence and contribution specifically, future investigations are warranted to identify the mechanisms contributing to the association between sleep and these two components of social well-being. As the reliability was low among each independent subscale of the social well-being measure utilized in the present study, study results should be interpreted with caution and its low intrascale reliability in mind. Further, although there is evidence of strong correlations between self- and informant-reported subjective well-being [35], valid self-reported measures of well-being variables should refer to specific time points rather than “life-as-a-whole” [36]. Future research that attempts to repeat or expand upon the present study may consider utilizing a measure of social well-being that is more reliable and better captures the multifaceted nature of this complex construct. Future investigations might also consider more specific social well-being assessments, perhaps via ecological momentary analysis [37]. 

The current study is limited in its generalizability to the general population. As the present sample was a smaller-sized sample composed of a majority of white, female-identifying, college-educated, and good sleepers as measured via the PSQI, our findings should be interpreted with caution. Further, due to the current sample being middle-aged on average, any attempt to draw conclusions about sleep and social well-being for younger adults should be made with caution. Future studies would benefit from utilizing a sample with increased age, race, gender, and socioeconomic diversity for a better understanding of how objective and subjectively measured sleep influences social well-being in a more representative sample.

## 7. Conclusions

The present study is an initial step in providing evidence for how sleep is tied to longer-term social functioning in middle-aged and older adults. The findings of this study suggest that better sleep quality is associated with greater social well-being. A greater amount of total sleep time was associated with poorer social well-being, which is worthy of further investigation. Further work should attempt to (1) identify the mechanisms that may explain the association between sleep indices and social well-being and (2) measure sleep and well-being over various time frames to further refine the temporal link between these important variables. 

## Figures and Tables

**Table 1 ijerph-19-11668-t001:** Frequency and percentage values of sample characteristics (age, sex, race, education, and PSQI sleep classification (poor sleepers = >6; good sleepers = <5)); *N* = 213.

Demographic Characteristics		Frequency	%
Age (*M*, *SD*, Range)	55.68, 10.92, 35–83		
Sex			
	Female	127	59.6%
	Male	86	40.4%
Race			
	White	203	95.3%
	Black and/or African American	2	0.9%
	Native American or Alaska Native	4	1.9%
	Asian	1	0.5%
	Other race identity	3	1.4%
Education			
	Eighth grade/Junior high school (7–8)	3	1.4%
	Some high school (9–12)	4	1.9%
	GED	2	0.9%
	High school graduate	49	23.0%
	1 to 2 years of college, no degree	36	16.9%
	3+ years of college, no degree	9	4.2%
	Graduated from 2-year college or Associate’s degree	11	5.2%
	Graduated from a 4- or 5-year college or Bachelor’s degree	49	23%
	Some graduate school	7	3.3%
	Master’s degree	36	16.9%
	Ph.D., Ed.D., M.D., DDS, or other professional degrees	7	3.3%
Income (USD; *M*, *SD*, Range)	USD 72,341.23, USD 54,553.51, USD 0–300,000.00		
PSQI Classification			
	Good sleepers	131	61.5%
	Poor sleepers	82	38.5%

**Table 2 ijerph-19-11668-t002:** Descriptive statistics and bivariate correlations of study variables.

	*M* (SD)	1.	2.	3.	4.	5.	6.	7.	8.	9.	10.
1. Age	55.68 (10.92)	--									
2. Income	72,341.23 (54,553.51)	−0.22 **	--								
3. CES-D	32.49 (3.45)	−0.21 **	0.03	--							
4. WatchSOL	22.76 (20.04)	0.00	0.02	0.04	--						
5. WatchSE	83.63 (7.30)	0.05	0.03	−0.10	−0.77 **	--					
6. WatchTST	389.77 (56.43)	0.04	−0.08	−0.15 *	−0.32 **	0.52 **	--				
7. DiarySQR	2.32 (0.67)	−0.08	−0.08	0.21 *	0.02	−0.08	−0.02	--			
8. DiarySOL	17.43 (113.05)	0.07	−0.05	0.05	0.19 **	−0.17 *	0.02	0.34 **	--		
9. SWB	67.18 (12.60)	−0.05	0.23 **	−.011	−0.07	0.02	−0.16 *	−0.26 **	−0.08	--	
10. PSQI	5.53 (3.14)	−0.03	−0.09	0.23 **	0.16 *	−0.22 **	−0.13	0.41 **	0.46 **	−0.18 **	--

*N* = 213; * *p* < 0.05. ** *p* < 0.001; CES-D = Center for Epidemiologic Studies Depression Scale (minus sleep item), Watch = actigraphically measured, Diary = diary-reported, SOL = sleep-onset latency, SE = sleep efficiency, TST = total sleep time, SQR= sleep quality rating, SWB = social well-being, PSQI = Pittsburgh Sleep Quality Index.

**Table 3 ijerph-19-11668-t003:** Regression analyses of actigraphic indices of sleep in relation to social well-being: N = 213; standardized regression coefficients (*β*); bold font indicates significance; CES-D = Center for Epidemiologic Studies Depression Scale (minus sleep item), TST = average total sleep time, SE = average sleep efficiency, SOL = average sleep-onset latency.

Predictor		Step 1			Step 2	
	Coeff.	*SE*	*p*	Coeff.	*SE*	*p*
Constant	79.06	10.34	**0.00**	104.74	19.69	**0.00**
Age	−0.03	0.08	0.70	−0.03	0.08	0.71
Sex	0.03	1.73	0.62	0.09	1.31	0.22
Income	0.22	0.00	**0.00**	0.21	0.00	**0.00**
CES-D	−0.12	0.25	0.08	−0.16	0.25	**0.03**
TST	--	--	--	−0.23	0.02	**0.00**
SE	--	--	--	−0.00	0.20	0.99
SOL	--	--	--	−0.12	0.07	0.26
*R* ^2^		0.07			0.11	
Δ*R*^2^		--			0.05	

**Table 4 ijerph-19-11668-t004:** Regression analyses of diary-reported indices of sleep in relation to social well-being: N = 213; standardized regression coefficients (*β*); bold font indicates significance; CES-D = Center for Epidemiologic Studies Depression Scale (minus sleep item), SOL = average sleep-onset latency, SQR = average sleep quality rating.

Predictor		Step 1			Step 2	
	Coeff.	*SE*	*p*	Coeff.	*SE*	*p*
Constant	79.88	10.34	**0.00**	85.82	10.249	**0.00**
Age	−0.01	0.08	0.92	−0.02	0.08	0.72
Sex	0.04	1.74	0.61	0.03	1.71	0.69
Income	0.21	0.00	**0.00**	0.19	0.00	0.01
CES-D	−0.14	0.25	0.05	−0.09	0.25	0.22
SOL	--	--	--	0.02	0.07	0.82
SQR	--	--	--	−0.24	1.37	**0.00**
*R* ^2^		0.07			0.12	
Δ*R*^2^		--			0.05	

**Table 5 ijerph-19-11668-t005:** Regression analyses of PSQI index of sleep in relation to social well-being; Step 2a uses PSQI continuous scale scores; CES-D = Center for Epidemiologic Studies Depression Scale (minus sleep item); Step 2b uses PSQI dichotomized (0 = good sleepers, 1 = poor sleepers); N = 213; standardized regression coefficients (*β*); bold font indicates significance.

Predictor		Step 1			Step 2	
	Coeff.	*SE*	*p*	Coeff.	*SE*	*p*
Constant	79.06	10.34	**0.00**	78.51	10.25	**0.00**
Age	−0.03	0.08	0.70	−0.03	0.08	0.70
Sex	0.03	1.73	0.62	0.04	1.72	0.56
Income	0.22	0.00	**0.00**	**0.21**	0.00	**0.00**
CES-D	−0.12	0.25	0.08	−0.09	0.26	0.21
Step 2a						
PSQI_Contin	--	--	--	−0.15	0.28	**0.03**
*R* ^2^		0.07			0.09	
Δ*R*^2^		--			0.02	
Step 2b						
PSQI_Dichot	--	--	--	−0.15	1.8	0.04
*R* ^2^		0.07			0.09	
Δ*R*^2^		--			0.02	

## Data Availability

MIDUS 2, Project 1 doi:10.3886/ICPSR04652; MIDUS 2, Project 4 doi:10.3886/ICPSR29282; MIDUS 3, Project 1 doi:10.3886/ICPSR36346.v3 (All MIDUS datasets are available at http://midus.colectica.org/).

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
