# Peer review of "Sleep and Social Wellness: Does Current Subjective and Objective Sleep Inform Future Social Well-Being?"

_ijerph, 2022, doi:10.3390/ijerph191811668_

Round 1
Reviewer 1 Report
There are a number of strengths of this manuscript. It addresses some limitations in the literature on sleep and well being. It uses longitudinal data to assess the associations between sleep and social well being. It investigates a significant question related to the role of sleep and will be of interest to readers. It uses self reported and objective measures of sleep. An additional strength is that it looks at the contributions of different aspects of the sleep experience such as sleep quality and sleep time. The statistical analysis is appropriate for the questions explored.
With this said, a limitation of the study is the low reliability of the sub scales of the social well being scale, their key dependent measure. The authors do address possible reasons for the low reliability of the sub scales (lines 189-192). With this said, the authors need to address the impact of this more fully in the discussion on the confidence they can hold in their results. Social well being is a complex and multifaceted phenomenon that may well be inadequately measured by this instrument. Hence, a suggestion for future research should also be to repeat the study using a more reliable measure of social well being.
An additional concern is with the relatively small sample size and the age of the respondents. The authors note that the mean age was about 56. Significant questions remain to be addressed about the role of sleep in the social wellbeing of younger adults. This should be noted more clearly in the discussion. Additionally, the majority of the sample were female. Future studies would benefit by using a larger sample of males to more effectively address the possibility of sex differences. This should be noted by the authors.
Reviewer 2 Report
I have read the article by Ghose et al. with great interest. The authors compared subjective and objective sleep indices with social well-being.
Comments:
· Table 1. Please, provide the list of chronic diseases. Most of them affect both well-being and sleep. The analyses should be hence adjusted for these.
· Actigraphy. Although I am not familiar with the particular equipment, but it would important to see most restful (L5) and most active (M10) time. This could reveal any socially inconvenient time of their sleep which often does strongly correlate with total sleep time (i.e. shift workers may have lower well-being despite having normal total sleep time).
· Statistical analyses. Could you please, provide power (presumed effect size) analyses to justify sample size?
· Table 3. Please, adjust the borders for the first column so that the text is not compromised.
· Discussion. Actigraphy is very difficult to interpret by itself without knowing the presence of primary sleep disorders which would obviously compromise the actigraphy data and social well being independently. Please, discuss the lack of data on primary sleep disorders in this cohort as a limitation.
Round 2
Reviewer 2 Report
I am happy with the comments and suggest acceptance